# RECURRENT DISTANCE-ENCODING NEURAL NETWORKS FOR GRAPH REPRESENTATION LEARNING

## ABSTRACT

Graph neural networks based on iterative one-hop message-passing have been shown to struggle in harnessing information from distant nodes effectively. Conversely, graph transformers allow each node to attend to all other nodes directly, but suffer from high computational complexity and have to rely on ad-hoc positional encodings to bake in the graph inductive bias. In this paper, we propose a new architecture to reconcile these challenges. Our approach stems from the recent breakthroughs in long-range modeling provided by deep state-space models on sequential data: for a given target node, our model aggregates nodes at different distances and uses a parallelizable linear recurrent network over the chain of distances to provide a natural encoding of its neighborhood structure. With no need for positional encoding, we empirically show that the performance of our model is competitive compared with that of state-of-the-art graph transformers on various benchmarks, at a drastically reduced computational complexity. In addition, we show that our model is theoretically more expressive than one-hop message-passing neural networks.

## 1 INTRODUCTION

Graphs are ubiquitous for representing complex interactions between individual entities, such as in social networks (Tang et al., 2009), recommender systems Ying et al. (2018) and molecules (Gilmer et al., 2017), and have thus attracted a lot of interest from researchers seeking to apply deep learning to graph data. Message passing neural networks (MPNNs) (Gilmer et al., 2017) have been the dominant approach in this field. These models iteratively update the representation of a target node by aggregating the representations of its neighbors. Despite progress on semi-supervised node classification tasks (Kipf & Welling, 2016; Veličković et al., 2017), MPNNs have been shown to have intrinsic limitations. Firstly, the expressive power of any message passing neural network is upper bounded by the Weisfeiler-Lehman graph isomorphism test (1-WL) (Xu et al., 2018). Moreover, to utilize the information from a node that is $k$ hops away from the target node, an MPNN needs to perform $k$ rounds of message passing. As a result, the receptive field for the target node grows exponentially with $k$, including many duplicates of nodes that are close to the target node. The information from an exponentially growing receptive field is compressed into a fixed-size representation, making it difficult to effectively harness the information of the distant nodes (a.k.a. over-squashing (Alon & Yahav, 2020; Topping et al., 2021)). These limitations may hinder the application of MPNNs to tasks that require reasoning between distant nodes.

Recently, inspired by the success of attention-based transformers in modeling natural language (Vaswani et al., 2017; Devlin et al., 2018) and images (Dosovitskiy et al., 2020), several works have adapted transformers for graph representation learning (Ying et al., 2021; Kim et al., 2022; Chen et al., 2022; Zhang et al., 2023). Graph transformers allow each node to attend to all other nodes directly through a global attention mechanism, and therefore make information flow between distant nodes easier. However, a naive implementation of a global attention mechanism alone doesn't encode any structural information about the underlying graph. As a result, state-of-the-art graph transformers rely on ad hoc positional encodings (e.g., eigenvectors of the graph Laplacian) as extra features to incorporate the graph inductive bias. There is no consensus yet on the optimal type of positional encodings, and what positional encoding to use is often a hyper-parameter that needs to be carefully tuned (Rampášek et al., 2022). Besides, while graph transformers have empirically

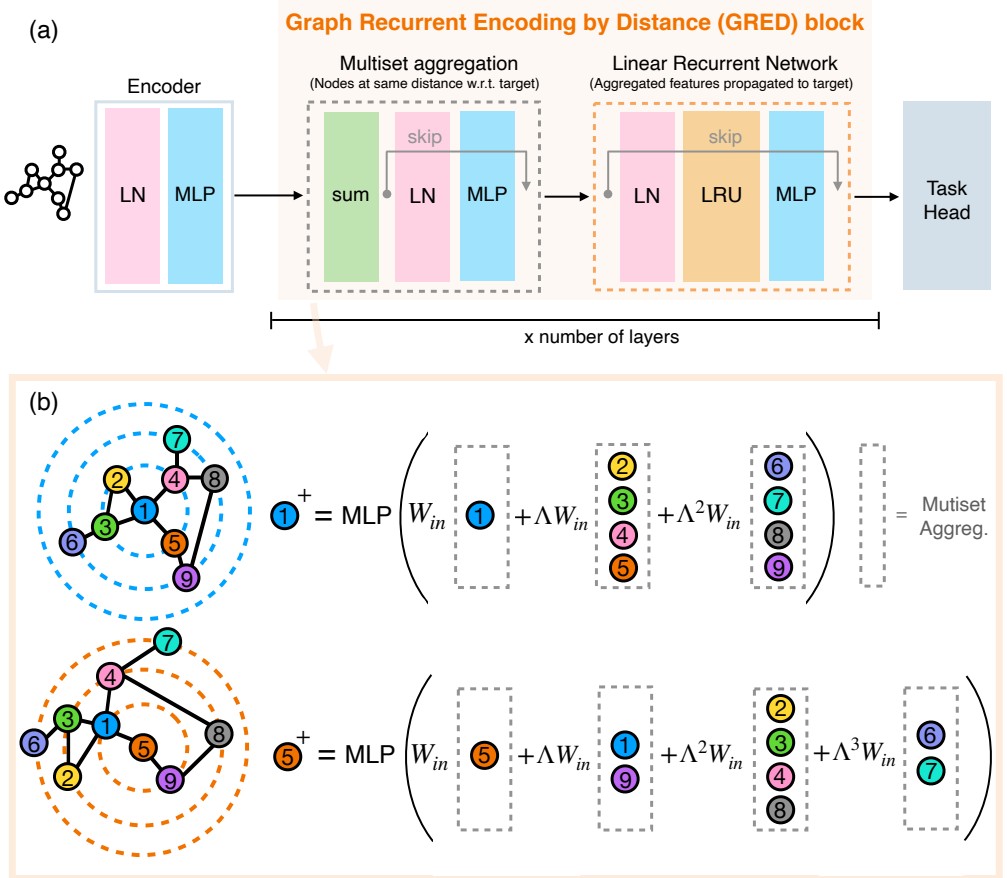

Figure 1: (a) Sketch of the architecture. MLPs and Layer Normalization operate independently at each node or node multiset. Information is propagated along edges through a linear RNN – specifically an LRU (Orvieto et al., 2023b). (b) Depiction of the block operation for two target nodes. The gray rectangular boxes indicate the application of multiset aggregation. Finally, the new representation for the target node is computed from the RNN output through an MLP.

shown improvement on some graph benchmarks compared with classical MPNNs, the former has a much higher computational complexity (Ying et al., 2021).

Captivated by the above challenges and the need for powerful, theoretically sound and computationally efficient approaches to graph representation learning, we propose a new model, Graph Recurrent Encoding by Distance (GRED). To learn the representation for a target node, our model categorizes all other nodes into multiple sets according to their shortest distances to the target node. The first component of a GRED layer is a permutation-invariant neural network (Zaheer et al., 2017) which generates a representation for each set of nodes that have the same shortest distance to the target node. The second component of a GRED layer is a linear recurrent neural network (Orvieto et al., 2023b) to encode the sequence of the set representations, starting from the set with the maximum shortest distance towards the target node. Since the order of the sequence is naturally encoded by the recurrent neural network, our model can encode the shortest distance of each set and thus the neighborhood structure of the target node. The architecture of GRED is illustrated in Figure 1.

The combination of a permutation-invariant neural network and a linear recurrent neural network brings several advantages to our model compared to existing approaches. First, the recurrent neural network allows the target node to effectively harness the information from distant nodes, and at the same time encodes the hierarchical structure of its neighborhood. As a result, our model doesn't require any positional encoding, unlike graph transformers. Second, the expressive power of the linear recurrent neural network strengthens that of the permutation-invariant neural network, making our model strictly more expressive than any one-hop message passing neural network (see Section 4 for detailed analysis). Third, both the linear recurrent neural network and the permutation-invariant neural network enable fast parallelizable computation, making our model significantly more efficient

than graph transformers. We evaluate our model on a series of graph benchmarks to support its efficacy. The performance of our model is consistently better than that of existing MPNNs and is also competitive compared with that of the state-of-the-art graph transformers, with drastically reduced computation time.

To summarize, the main contributions of our paper are as follows:

1. We propose a principled new architecture for graph representation learning to more effectively and efficiently utilize the information of large neighborhoods. The proposed architecture consists of linear recurrent neural networks interleaved with permutation-invariant neural networks.
2. We theoretically prove the expressiveness of the combination of permutation-invariant neural networks and linear recurrent neural networks, which is strictly greater than that of any one-hop MPNN and aligned with that of $k$-hop MPNNs, while using fewer parameters.
3. We empirically show that the performance of our model is significantly better than that of message passing neural networks, and comparable to that of state-of-the-art graph transformers, with greatly reduced computational time.

## 2 RELATED WORK

We review below recent literature on increasing the receptive field of MPNNs as well as current trends in recurrent models for long-range reasoning on sequential data.

**Increasing the receptive field of MPNNs.** There is a rich literature on using higher hop information for MPNNs. Among them, MixHop (Abu-El-Haija et al., 2019) uses powers of the normalized adjacency matrix to access $k$-hop nodes. $k$-hop GNN (Nikolentzos et al., 2020) iteratively applies MLPs to combine two consecutive hops and propagates information towards the target node. While they have shown higher hop information can improve the expressiveness of MPNNs, they have the problem of over-squashing (Topping et al., 2021) due to neighborhood mixing. SPN (Abboud et al., 2022) first aggregates nodes of the same hop and then combines hop representations using weighted summation. Although SPN can alleviate over-squashing empirically, weighted summation of hops cannot guarantee the expressiveness of the model. Graph transformers (Ying et al., 2021; Chen et al., 2022; Rampášek et al., 2022; Zhang et al., 2023; Wu et al., 2021) have attracted a lot of attention recently because the global attention mechanism allows each node to directly attend to all other nodes. To bake in the inductive bias of the underlying graph structure, graph transformers typically use positional encodings (Li et al., 2020; Dwivedi et al., 2021) as extra node features. Among them, Graphormer (Ying et al., 2021) adds learnable bias to the attention matrix to make the model aware of nodes with different shortest distances. However, the sequential order of the distances is not encoded into the model and Graphormer also needs local node degrees as extra input node features. SAT (Chen et al., 2022) and GraphTrans (Wu et al., 2021) stack message passing layers and self-attention layers together to obtain local information before the global attention. GPS (Rampášek et al., 2022) applies the linear attention (Choromanski et al., 2020) to graph transformers and empirically investigates different configurations of positional encoding. Zhang et al. (2023) theoretically prove the expressive power of different distance encodings regarding biconnectivity and proposes to use Resistance Distance as a kind of relative positional encoding. GRIT (Ma et al., 2023) utilizes learnable positional encodings initialized with random walk probabilities.

**Issues with attention for long-range reasoning in sequential data.** Efficient processing of long sequences is one of the paramount challenges in contemporary deep learning. Attention-based transformers (Vaswani et al., 2017) provide a scalable approach to sequential modeling but suffer from *quadratically increasing inference/memory complexity* as the sequence length grows. While many approaches exist to alleviate this issue, efficient memory management (Dao et al., 2022; Dao, 2023) and architectural modifications (Wang et al., 2020; Kitaev et al., 2020; Child et al., 2019; Beltagy et al., 2020; Wu et al., 2020), the sequence length in modern large language models is usually kept to $2k/4k$ tokens for this reason (e.g. Llama2 (Touvron et al., 2023)). On top of high inference and memory cost, the attention mechanism often does not provide the correct *inductive bias* for long-range reasoning beyond text. Indeed, most transformers (including long-range/sparse variants, reduced complexity variants, or variants with other tricks) are often found to perform poorly in discovering long-range dependencies in data (Tay et al., 2020). Due to the issues outlined above, the community has witnessed in the last year the rise of new, drastically innovative, *recurrent* al-

ternatives to the attention mechanism, named state-space models (SSMs). The first SSM, S4, was introduced by Gu et al. (2021) based on the theory of polynomial signal approximations (Gu et al., 2020; 2023), and since then, a plethora of variants have been proposed (Hasani et al., 2022; Gupta et al., 2022; Gu et al., 2022; Smith et al., 2022; Peng et al., 2023). These models achieve remarkable performance, surpassing all modern attention-based transformer variants by an average 20% accuracy on challenging sequence classification tasks (Tay et al., 2020). Deep state-space models have reached outstanding results in various domains, including vision (Nguyen et al., 2022), audio (Goel et al., 2022), biological signals (Gu et al., 2021), reinforcement learning (Lu et al., 2023) and online learning (Zucchet et al., 2023b). SSMs also were successfully applied to language modeling and are sometimes used in combination with attention (Fu et al., 2023; Wang et al., 2023; Ma et al., 2022). At inference time, all SSMs coincide with a stack of linear Recurrent Neural Networks (RNNs), interleaved with position-wise MLPs and normalization layers. Such combination was recently shown to have fully expressiveness in modeling nonlinear dynamical systems (Orvieto et al., 2023a) – no recurrent nonlinearities are needed, since an MLP is placed at the output (cf. LSTMs (Hochreiter & Schmidhuber, 1997), GRUs (Chung et al., 2014)). Most importantly, the linearity of the RNNs allows for fast parallel processing using FFTs (Gu et al., 2022) or parallel scans (Smith et al., 2023).

**Linear recurrent unit.** Among modern architectures for long-range reasoning based on recurrent modules, the simplest is perhaps Linear Recurrent Unit (LRU) (Orvieto et al., 2023b): while SSMs rely on the discretization of a structured continuous-time latent dynamical system, the LRU is directly designed for discrete-time systems (token sequences), and combines easy hyperparameter tuning with solid performance and scalability. The only difference between the LRU and the standard RNN update $s_k = As_{k-1} + Bx_k$ ($x$ is the input at a specific layer and $s$ is the hidden-state, then fed into a position-wise MLP) is (1) the system operates in the complex domain (required for expressivity, see discussion in Orvieto et al. (2023b)) (2) to enhance stability and prevent vanishing gradients, $A$ (diagonal) is learned using polar parametrization and log-transformed magnitude and phase. Finally, (3) the recurrence is normalized through an extra optimizable parameter that scales the input to stabilize signal propagation. The parametrization of linear RNNs of (Orvieto et al., 2023b) was found to be effective also in surpassing deep LSTMs and GRUs (Zucchet et al., 2023a).

## 3 ARCHITECTURE

In this section, we present the GRED block, which is the building unit of our architecture. We start with some preliminary notations and then describe how our block computes node representations. Finally, we analyze its computational complexity. We defer the discussion on expressive power of our module to Section 4.

**Preliminaries.** Let $G = (V, E)$ denote an undirected and unweighted graph, where $V$ denotes the set of nodes with $|V| = N$, and $E$ denotes the set of edges. For any two nodes $v, u \in V$, we use $d(v, u)$ to represent the shortest distance between $v$ and $u$, and we let $d(v, v) = 0$. For each target node $v$, we categorize all other nodes into different hops according to their shortest distances to $v$:

$$\mathcal{N}_k(v) = \{u \mid d(v, u) = k\} \quad \text{for} \quad k = 0, 1, \ldots, K \tag{1}$$

where $K$ can be the diameter of $G$ or a hyper-parameter specified for the task in hand. $\{\mathcal{N}_k(v)\}_{k=1}^K$ can be obtained for each node $v \in V$ by running the Floyd–Warshall algorithm (Floyd, 1962; Warshall, 1962) in parallel once during data pre-processing.

**Graph Recurrent Encoding by Distance (GRED).** The input to the $\ell$-th block is the set of node representations $\{h_v^{(\ell-1)} \in \mathbb{R}^d \mid v \in V\}$. To compute the output representation $h_v^{(\ell)}$ for a generic target node $v$, the block first generates a representation for each set of nodes that have the same shortest distances to $v$ (grey dashed box in Figure 1):

$$x_{v,k}^{(\ell)} = \text{AGG}\left(\{\!\{h_u^{(\ell-1)} \mid u \in \mathcal{N}_k(v)\}\!\}\right) \quad \text{for} \quad k = 0, 1, \ldots, K \tag{2}$$

where AGG is an *injective* multiset function (Zaheer et al., 2017; Xu et al., 2018), which we parametrize, as usual in the literature (Zaheer et al., 2017; Xu et al., 2018; Feng et al., 2022), by

two wide multi-layer perceptrons (MLPs)[1]:

$$\boldsymbol{x}_{v,k}^{(\ell)} = \text{MLP}_2 \left( \sum_{u \in \mathcal{N}_k(v)} \text{MLP}_1 \left( \boldsymbol{h}_u^{(\ell-1)} \right) \right). \tag{3}$$

The sequence of the set representations $(\boldsymbol{x}_{v,0}^{(\ell)}, \boldsymbol{x}_{v,1}^{(\ell)}, \dots, \boldsymbol{x}_{v,K}^{(\ell)})$ is then encoded by a linear recurrent network (RNN), starting from the set of nodes $\boldsymbol{x}_{v,K}^{(\ell)}$ which is the farthest away from the target node to the target node itself[2], which gives:

$$\boldsymbol{s}_{v,k}^{(\ell)} = \boldsymbol{A}\boldsymbol{s}_{v,k-1}^{(\ell)} + \boldsymbol{B}\boldsymbol{x}_{v,K-k}^{(\ell)} \quad \text{for} \quad k = 0, 1, \dots, K \tag{4}$$

where $\boldsymbol{s}_{v,k}^{(\ell)} \in \mathbb{R}^{d_s}$ represents the hidden state of the recurrent neural network and $\boldsymbol{s}_{v,-1}^{(\ell)} = \boldsymbol{0}$. $\boldsymbol{A} \in \mathbb{R}^{d_s \times d_s}$ denotes the state transition matrix and $\boldsymbol{B} \in \mathbb{R}^{d_s \times d}$ is a matrix to transform the input of the RNN. It is well-known that wide enough linear RNNs can parameterize any convolutional filter over arbitrarily long input sequences (Li et al., 2022). Moreover, it was recently shown that, when interleaved with MLPs (placed at the recurrence output), a stack of linear RNNs can actually model any non-linear dynamical system (Orvieto et al., 2023a).

An important advantage of linear recurrences, crucial for computational speed-up over sequential processing, is that they can be represented without loss of generality in diagonal complex form (see discussion in (Smith et al., 2023; Orvieto et al., 2023b)). Recall that, over the space of $d_s \times d_s$ non-diagonal real matrices, the set of non-diagonalizable (in the complex domain) matrices has measure zero (Bhatia, 2013). Hence, with probability one over random initialization, $\boldsymbol{A}$ is diagonalizable, i.e. $\boldsymbol{A} = \boldsymbol{V}\boldsymbol{\Lambda}\boldsymbol{V}^{-1}$, where $\boldsymbol{\Lambda} = \text{diag}(\lambda_1, \dots, \lambda_{d_s}) \in \mathbb{C}^{d_s \times d_s}$ gathers the eigenvalues of $\boldsymbol{A}$, and columns of $\boldsymbol{V}$ are the corresponding eigenvectors. Eq. (4) is equivalent to the following diagonal complex recurrence, up to a linear transformation of the hidden state $\boldsymbol{s}$ which can be merged with the output projection $\boldsymbol{W}_{\text{out}}$ (c.f. eq. (6)).

$$\boldsymbol{s}_{v,k}^{(\ell)} = \boldsymbol{\Lambda}\boldsymbol{s}_{v,k-1}^{(\ell)} + \boldsymbol{W}_{\text{in}}\boldsymbol{x}_{v,K-k}^{(\ell)} \tag{5}$$

where $\boldsymbol{W}_{\text{in}} = \boldsymbol{V}^{-1}\boldsymbol{B} \in \mathbb{C}^{d_s \times d}$. Eq. (5) can be thought of as a *filter over the hops from the target node*, and the magnitudes of the eigenvalues $\boldsymbol{\Lambda}$ control how fast the filter decays as the shortest distance from the target node increases. Following the modern literature on long-range reasoning, we directly initialize (without loss in generality) the system in diagonal form (Gupta et al., 2022; Gu et al., 2022), and train[3] $\boldsymbol{\Lambda}$ and $\boldsymbol{W}_{\text{in}}$. To guarantee stability (eigenvalues in the unit disk), increased resolution at $|\lambda| \approx 1$ and strong signal propagation from distant nodes, we adopt the recently introduced LRU initialization and parametrization (Orvieto et al., 2023b), which also leverages parallel scans (Blelloch, 1990; Smith et al., 2023) to avoid computing $\boldsymbol{s}$ sequentially on modern hardware.

The output representation $\boldsymbol{h}_v^{(l)}$ is generated by a non-linear transformation of the hidden state $\boldsymbol{s}_{v,K}^{(\ell)}$:

$$\boldsymbol{h}_v^{(\ell)} = \text{MLP} \left( \Re \left[ \boldsymbol{W}_{\text{out}}\boldsymbol{s}_{v,K}^{(\ell)} \right] \right) \tag{6}$$

where $\boldsymbol{W}_{\text{out}} \in \mathbb{C}^{d \times d_s}$ is a trainable weight matrix and $\Re[\cdot]$ denotes the real part of a complex-valued vector. While sufficiently wide MLPs with one hidden layer can parametrize any non-linear map, following again the literature on state-space models we choose to place here a gated linear unit (GLU, Dauphin et al. (2017)): $\text{GLU}(x) = (\boldsymbol{W}_1\boldsymbol{x}) \odot \sigma(\boldsymbol{W}_2\boldsymbol{x})$, with $\sigma$ the sigmoid function.

The final architecture is composed of stacking several of such blocks described above. In practice, we merge $\text{MLP}_1$ in Eq. (3) with the non-linear transformation in Eq. (6) at the previous layer (or at the encoding layer) to make the entire architecture more compact. We apply layer normalization to the input of both the MLP and the linear RNN, and also add skip connections.

---

[1]In practice, with just one hidden layer.

[2]Standard RNNs on sequences would have input $\boldsymbol{x}_k$ and not $\boldsymbol{x}_{K-k}$. Here propagation starts from the farthest away node, and proceeds right-to-left as opposed to left-to-right, ending at the target node.

[3]As done in all state-space models Gu et al. (2021); Smith et al. (2023), we do not optimize over the complex numbers but instead parameterize, for instance, real and imaginary components of $\boldsymbol{B}$ as real parameters. The imaginary unit $i$ is then used to aggregate the two components in the forward pass.

**Computational complexity** For each shortest distance $k$, the complexity of aggregating representations of nodes from $\mathcal{N}_k(v)$ for every $v \in V$ is at most that of one round of message passing, which is $O(|E|)$. So the complexity of the first part of the computation of our block (Eq. (3)) is less than $O(K|E|)$. In practice, since $\{\mathcal{N}_k(v)\}_{k=1}^K$ are pre-computed, computing Eq. (3) for every $k$ can be parallelized. The sequential encoding of Eq. (5) has total complexity $O(K|V|)$. However, the linearity of the recurrence and the diagonal state transition matrix enable a parallel scan over the sequences, which further speeds up the computation. As a result, our model is highly efficient during training, as evidenced by our experimental results.

## 4 EXPRESSIVENESS ANALYSIS

We briefly recap the computation performed by a single GRED layer to update node representations $\{\boldsymbol{h}_u\}_{u \in V}$ (obtained via a previous layer or encoder) in some graph $V$. For each node $v \in V$ and distance $k$, we compute the $k$-hop neighbors $\mathcal{N}_k(v)$, and aggregate their features using an injective multiset function AGG. The result is a set of node-dependent sequences $\{(\boldsymbol{x}_{v,k})_k^{K_v} : v \in V\}$ of different lengths $K_v$, such that $\boldsymbol{x}_{v,k} = \mathrm{AGG}(\{\!\{\boldsymbol{h}_u \mid u \in \mathcal{N}_k(v)\}\!\})$. All these sequences are then processed by a linear RNN, which we name here $R$ for convenience, and injectively mapped to the updated features for each node $v$ through a non-linear transformation.

### 4.1 PROPERTIES OF RNN-BASED FILTERING OF $k$-HOP NEIGHBORS FEATURES

Up to zero-padding, linear RNNs can be seen as maps $R : (\boldsymbol{x}_{v,0}, \boldsymbol{x}_{v,1}, \boldsymbol{x}_{v,2}, \ldots, \boldsymbol{x}_{v,K}) \mapsto \boldsymbol{s}_{v,K}$, where $v \in V$ is a generic node and $\boldsymbol{s}_{v,K}$ is the last hidden state of the recurrence $\boldsymbol{s}_{v,k} = \boldsymbol{\Lambda}\boldsymbol{s}_{v,k-1} + \boldsymbol{W}_{\mathrm{in}}\boldsymbol{x}_{v,K-k}$, with $\boldsymbol{s}_{-1} = \boldsymbol{0}$. Unrolling the recurrence:

$$
\begin{aligned}
\boldsymbol{s}_{v,0} &= \boldsymbol{W}_{\mathrm{in}}\boldsymbol{x}_{v,K} \\
\boldsymbol{s}_{v,1} &= \boldsymbol{\Lambda}\boldsymbol{W}_{\mathrm{in}}\boldsymbol{x}_{v,K} + \boldsymbol{W}_{\mathrm{in}}\boldsymbol{x}_{v,K-1} \\
\boldsymbol{s}_{v,2} &= \boldsymbol{\Lambda}^2\boldsymbol{W}_{\mathrm{in}}\boldsymbol{x}_{v,K} + \boldsymbol{\Lambda}\boldsymbol{W}_{\mathrm{in}}\boldsymbol{x}_{v,K-1} + \boldsymbol{W}_{\mathrm{in}}\boldsymbol{x}_{v,K-2} \\
&\cdots \\
\boldsymbol{s}_{v,K} &= \boldsymbol{\Lambda}^K\boldsymbol{W}_{\mathrm{in}}\boldsymbol{x}_{v,K} + \boldsymbol{\Lambda}^{K-1}\boldsymbol{W}_{\mathrm{in}}\boldsymbol{x}_{v,K-1} + \cdots + \boldsymbol{\Lambda}\boldsymbol{W}_{\mathrm{in}}\boldsymbol{x}_{v,1} + \boldsymbol{W}_{\mathrm{in}}\boldsymbol{x}_{v,0},
\end{aligned}
\tag{7}
$$

where $\boldsymbol{x}_{v,k} = \mathrm{AGG}(\{\!\{\boldsymbol{h}_u \mid u \in \mathcal{N}_k(v)\}\!\})$. Hence, we have that $\boldsymbol{s}_{v,K} = \sum_{k=0}^K \boldsymbol{\Lambda}^k\boldsymbol{W}_{\mathrm{in}}\boldsymbol{x}_k$: aggregated features of nodes distant from the target $v$ are scaled by a large power of $\boldsymbol{\Lambda}$, while the target node features and the one-hop neighbor features are scaled by $\boldsymbol{I}$ and $\boldsymbol{\Lambda}$, respectively. Under the LRU parametrization (see appendix), $\boldsymbol{\Lambda}$ is initialized and trained to keep entries inside the unit disk (for stability), hence for large $\boldsymbol{\Lambda}^k$ will induce a decay structure (*low-pass filter*) over distance. In the limiting case of vanishing $\boldsymbol{\Lambda}$, the RNN just linearly transforms the features of each node: $\boldsymbol{s}_{v,K} = \boldsymbol{W}_{\mathrm{in}}\boldsymbol{x}_{v,0} = \boldsymbol{W}_{\mathrm{in}}\mathrm{AGG}(\{\!\{\boldsymbol{h}_v\}\!\})$.

A surprising feature of linear recurrences is that if the hidden state is large enough, *they are injective* — crucial property for expressiveness.

**Lemma 4.1** (Injectivity of Linear RNNs). *Let $\{\boldsymbol{x}_v, v \in V : \boldsymbol{x}_v = (\boldsymbol{x}_{v,0}, \boldsymbol{x}_{v,1}, \boldsymbol{x}_{v,2}, \ldots, \boldsymbol{x}_{v,K_v})\}$ be a set of sequences (with different lengths $K_v \leq K$) of vectors in a (possibly uncountable) set of features $\mathcal{X} \subset \mathbb{R}^d$. Consider a diagonal linear complex-valued RNN with $d_s$-dimensional hidden state, parameters $\boldsymbol{\Lambda} \in diag(\mathbb{C}^{d_s})$, $\boldsymbol{W}_{in} \in \mathbb{C}^{d \times d_s}$ and recurrence rule $\boldsymbol{s}_{v,k} = \boldsymbol{\Lambda}\boldsymbol{s}_{v,k-1} + \boldsymbol{W}_{in}\boldsymbol{x}_{v,K_v-k}$, initialized at $\boldsymbol{s}_{v,-1} = \boldsymbol{0} \in \mathbb{R}^{d_s}$ for all $v \in V$. If $d_s \geq Kd$, then there exist $\boldsymbol{\Lambda}, \boldsymbol{W}_{in}$ such that the map $R : (\boldsymbol{x}_{v,0}, \boldsymbol{x}_{v,1}, \boldsymbol{x}_{v,2}, \ldots, \boldsymbol{x}_{v,K}) \mapsto \boldsymbol{s}_{v,K}$ (with zero right-padding if $K_v < K$) is bijective. Moreover, if the set of RNN inputs has countable cardinality $|\mathcal{X}| = N \leq \infty$, then selecting $d_s \geq d$ is sufficient for the existence of an injective linear RNN mapping $R$.*

The proof can be found in the appendix, and leverages the representation $\boldsymbol{s}_{v,K} = \sum_{k=0}^K \boldsymbol{\Lambda}^k\boldsymbol{W}_{\mathrm{in}}\boldsymbol{x}_k$ and techniques from Orvieto et al. (2023a). Intuitively, this property guarantees that linear RNNs provide representations that retain feature and distance information[4] from $k$-hop neighbors.

---

[4] A careful reader might have realized that the result assumes zero-padding for node sequences shorter than the diameter. This implies that, in principle, our block cannot differentiate between nodes with vanishing features and virtual nodes. This issue can be easily fixed by replacing zero-padding with padding with a

## 4.2 GRED EXPRESSIVENESS AND COMPARISON WITH GENERAL $K$-HOP MESSAGE PASSING

The findings in the last subsection directly imply the following corollary.

**Corollary 4.2** (Information kept by GRED). *Assuming wide enough architectural components, then the RNN output at any node $v \in V$, in combination with an injective multiset function AGG aggregating neighbors, is an injective function of the list $(\boldsymbol{h}_v, \{\!\!\{ \boldsymbol{h}_u \mid u \in \mathcal{N}_1(v) \}\!\!\}, \{\!\!\{ \boldsymbol{h}_u \mid u \in \mathcal{N}_2(v) \}\!\!\}, \dots, \{\!\!\{ \boldsymbol{h}_u \mid u \in \mathcal{N}_{K_v}(v) \}\!\!\})$. That is, GRED provides an updated representation for each node $v$, gathering all nodes connected to $v$ through a path and keeping distance information.*

The corollary, in turn, implies the following result:

**Theorem 4.3.** *One wide enough GRED layer (linear RNN and AGG injective) is strictly more powerful one iteration of any 1-hop message passing algorithm.*

*Proof.* The linear RNN output at a target node $v$ is an injective function with arguments the outputs of injective multiset aggregation at different distances from $v$ (Cor. 4.2). This means, in particular, that the output changes at any perturbation of the 1-hop neighbor features. However, GRED can also sometimes recognize if graphs are isomorphic using structural information beyond 1-hop, see e.g. Figure 2. □

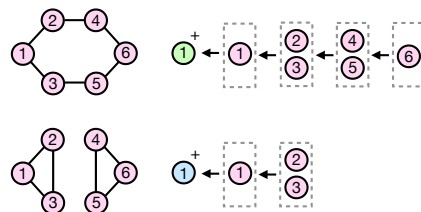

Figure 2: GRED provides distinct updates for the two graphs above. Such graphs, however, are indistinguishable under the 1-WL isomorphism test, assuming (worst-case) nodes features are identical.

It is well known that the power of 1-hop message passing is bounded by the bounded by the Weisfeiler-Lehman (1-WL) test (Xu et al., 2018). It is, therefore, interesting to ask if GRED's power can be aligned with an higher-order WL test. This is not the case, since GRED can be thought of as a specific $K$-hop *proper* message passing method.

**Definition 1** (Feng et al. (2022)). *A K-hop message passing algorithm [5] is a method that provides an update $\boldsymbol{h}_v \mapsto \boldsymbol{h}_v^+$, for all $v \in V$, as follows:*

$$\boldsymbol{x}_{v,k} = AGG_i(\{\!\!\{ \boldsymbol{h}_u | u \in \mathcal{N}_k(v)) \}\!\!\}), \quad \boldsymbol{h}_v^k = UPD_k(\boldsymbol{x}_{v,k}, \boldsymbol{h}_v), \quad \boldsymbol{h}_v^+ = CMB(\{\!\!\{ \boldsymbol{h}_v^k | k = 1, 2, ..., K \}\!\!\}),$$

*Further, a K-hop algorithm is called **proper** if the aggregations $(AGG_k)_{k=1}^K$, updates $(UPD_k)_{k=1}^K$, and combine (CMB) functions are all injective given input from a countable space.*

Corollary 4.2 shows that GRED provides a very convenient implementation of *proper* (i.e, injective) $K$-hop message passing with no need for a set of distance-specific injective multiset aggregation functions $\{AGG_k, UPD_k | k = 1, 2, ..., K\}$ — drastically reducing the number of network parameters. Further, to study the limitations of GRED, we can then rely on the following result.

**Theorem 4.4** (Bound for $K$-hop Feng et al. (2022)). *The expressive power of a proper K-hop message passing GNN of any kernel is bounded by the 3-WL test.*

Therefore, the expressive power of GRED lies in between the 1-WL and the 3-WL test — and is aligned to that of $K$-hop message passing where the maximum number of considered hops equals the graph diameter. As such, especially for large graphs, our approach provides a competitive alternative to general $K$-hop message passing, offering also aggregation of distant nodes through parallel scans (see also Tb. 4).

## 5 EXPERIMENTS

In this section, we evaluate our model on a series of graph benchmarks (Dwivedi et al., 2023; 2022). We compare against popular MPNNs including GCN (Kipf & Welling, 2016), GAT (Veličković et al., 2017), GIN (Xu et al., 2018), GatedGCN (Bresson & Laurent, 2017) and PNA (Corso et al., 2020), as well as state-of-the-art graph transformers: Graphormer (Ying et al., 2021), EGT (Hussain

---

specifically designed token that is not in the dictionary $\mathcal{X}$. In practice, such operation is not necessary since node features are first fed into an encoder with bias, which can then learn to shift them away from zero.

[5]We consider $K$-hop algorithms considering the shortest path distance kernel, see Feng et al. (2022).

Table 1: Test classification accuracy (in percent) of our model and baselines. Performance of baselines is from their original papers (Dwivedi et al., 2023; Rampášek et al., 2022; Ma et al., 2023). "-" indicates the baseline didn't report its performance on that dataset. # params ≈ 500K.

| Method | Model | MNIST | CIFAR10 | PATTERN | CLUSTER |
|---|---|---|---|---|---|
| MPNNs | GCN | 90.705±0.218 | 55.710±0.381 | **85.614±0.032** | 69.026±1.372 |
| | GAT | 95.535±0.205 | 64.223±0.455 | 78.271±0.186 | 70.587±0.447 |
| | GIN | 96.485±0.252 | 55.255±1.527 | 85.590±0.011 | 64.716±1.553 |
| | GatedGCN | **97.340±0.143** | **67.312±0.311** | 85.568±0.088 | **73.840±0.326** |
| GTs | EGT | **98.173±0.087** | 68.702±0.409 | 86.821±0.020 | 79.232±0.348 |
| | SAN | - | - | 86.581±0.037 | 76.691±0.65 |
| | SAT | - | - | 86.848±0.037 | 77.856±0.104 |
| | GPS | 98.051±0.126 | 72.298±0.356 | 86.685±0.059 | 78.016±0.180 |
| | GRIT | 98.108±0.111 | **76.468±0.881** | **87.196±0.076** | **80.026±0.277** |
| GRED (Ours) | | **98.195±0.090** | **75.370±0.621** | **86.759±0.020** | **78.495±0.103** |

et al., 2022), SAT (Chen et al., 2022), GPS (Rampášek et al., 2022) and GRIT (Ma et al., 2023). We also compare the training efficiency of our model and that of graph transformers to demonstrate the high efficiency of our model. We use three distinct colors to indicate our model, the best graph transformer and the best MPNN. We detail hyper-parameters we use in Appendix C.

**Benchmarking GNNs**   We first evaluate our model on the node classification datasets: PATTERN and CLUSTER, and graph classification datasets: MNIST and CIFAR10 from Dwivedi et al. (2023). To get the representation for the entire graph, we simply do average pooling over all node representations. We train our model four times with different random seeds and report the average accuracy with standard deviation. The comparison with baselines is shown in Table 1. From the table we see that graph transformers generally perform better than MPNNs. Among the four datasets, PATTERN models communities in social networks and all nodes are reachable within 3 hops, which we conjecture is the reason for the marginal performance gap between graph transformers and MPNNs. For a more difficult task, like CIFAR10, that requires information from a relatively larger neighborhood, graph transformers work more effectively. GRED performs well on all four datasets and consistently outperforms MPNNs. Especially, on MNIST GRED achieves the best accuracy, and on CIFAR10 the accuracy of GRED is comparable to GRIT and better than the other graph transformers, which validates that our model can effectively aggregate information beyond the local neighborhood. Additional ablation studies are shown in Appendix D.

Table 2: Test MAE on ZINC 12K. # params ≈ 500K.

| Model | Test MAE ↓ |
|---|---|
| GCN | 0.278 ± 0.003 |
| GAT | 0.384 ± 0.007 |
| GIN | 0.387 ± 0.015 |
| GatedGCN | 0.282 ± 0.015 |
| PNA (Corso et al., 2020) | **0.188 ± 0.004** |
| SAN (Kreuzer et al., 2021) | 0.139 ± 0.006 |
| Graphormer (Ying et al., 2021) | 0.122 ± 0.006 |
| K-subgraph SAT (Chen et al., 2022) | 0.094 ± 0.008 |
| KP-GIN (Feng et al., 2022) | 0.093 ± 0.007 |
| GPS (Rampášek et al., 2022) | 0.070 ± 0.004 |
| PathNN (Michel et al., 2023) | 0.090 ± 0.004 |
| GRIT (Ma et al., 2023) | **0.059 ± 0.002** |
| GRED (Ours) | **0.089 ± 0.004** |

**Performance on ZINC 12K**   Next, we report the test MAE of our model on ZINC 12K (Dwivedi et al., 2023). The average and standard deviation of four runs with different random seeds are

shown in Table 2 along with baseline performance from their original papers. From Table 2 we see that the performance of our model is significantly better than that of MPNNs. Although our performance is worse than the state-of-the-art graph transformer GRIT, our model outperforms some other graph transformer variants (SAN, Graphormer and K-subgraph SAT). This is impressive given that our model doesn't use any positional encoding, and provides evidence that our model is able to effectively encode graph structural information through the natural inductive bias of recurrence over distances.

Table 3: Test performance on Peptides-func/struct.

| Model | Peptides-func Test AP ↑ | Peptides-struct Test MAE ↓ |
|---|---|---|
| GCN | 0.5930±0.0023 | 0.3496±0.0013 |
| GINE | 0.5498±0.0079 | 0.3547±0.0045 |
| GatedGCN | 0.5864±0.0077 | 0.3420±0.0013 |
| GatedGCN+RWSE | 0.6069±0.0035 | 0.3357±0.0006 |
| Transformer+LapPE | 0.6326±0.0126 | 0.2529±0.0016 |
| SAN+LapPE | 0.6384±0.0121 | 0.2683±0.0043 |
| GPS | 0.6535±0.0041 | 0.2500±0.0005 |
| GRIT | 0.6988±0.0082 | 0.2460±0.0012 |
| GRED (Ours) | 0.7041±0.0049 | 0.2584±0.0015 |

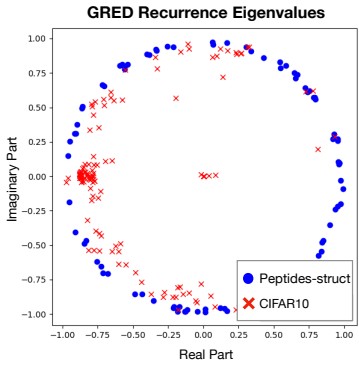

Figure 3: Learned eigenvalues of the first GRED layer: CIFAR10 and Peptides-struct.

**Long range graph benchmark**    To test the long range modeling capacity of our model, we evaluate it on the Peptides-func and Peptides-struct datasets from Dwivedi et al. (2022). We follow the 500K parameter budget and train our model four times with different random seeds. The results are shown in Table 3. We observe that GRED significantly outperforms all MPNN baselines. On Peptides-func, GRED even outperforms the best graph transformer GRIT, with no need for positional encoding. GRED's performance on Peptides-struct is also competitive compared with graph transformers. To illustrate how GRED can learn to encode the graph structure, we examine the eigenvalues of the linear recurrent neural network after training, as shown in Figure 3. We observe from the figure that the eigenvalues are pushed close to 1 for the long-range task Peptides-struct, which prevent distant information from decaying too quickly. Compared with Peptides-struct, CIFAR10 requires the model to utilize more information from the local neighborhood, so the magnitudes of the eigenvalues become smaller.

Table 4: Average training time per epoch and GPU memory consumption of GRIT and GRED.

| Model | ZINC 12K | CIFAR10 | Peptides-func |
|---|---|---|---|
| GRIT | 25.6s / 1.9GB | 244.4s / 4.6GB | 225.6s / 22.5GB |
| GRED (Ours) | 4.1s / 1.4GB | 27.8s / 1.4GB | 158.9s / 18.5GB |
| Speedup | 6.2× | 8.8× | 1.4× |

**Training efficiency**    Finally, to demonstrate the high efficiency of our model, we record the average training time per epoch and GPU memory consumption on ZINC, CIFAR10 and Peptides-func. We compare our measurements with those of the state-of-the-art graph transformer GRIT, shown in Table 4. Both models are trained using a single RTX A5000 GPU with 24GB memory.

## 6 CONCLUSION

In this paper, we introduced the Graph Recurrent Encoding by Distance (GRED) model for graph representation learning. By integrating permutation-invariant neural networks with linear recurrent neural networks, GRED effectively harnesses distant node information without the need for positional encodings or computationally expensive attention mechanisms. Theoretical and empirical evaluations confirm GRED's superior performance compared to existing MPNNs, and competitive results with state-of-the-art graph transformers at a significantly reduced computation time. This positions GRED as a powerful, efficient, and promising tool for graph representation learning.

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

# A    PROOFS

**Lemma 4.1** (Injectivity of Linear RNNs). *Let $\{\boldsymbol{x}_v, v \in V : \boldsymbol{x}_v = (\boldsymbol{x}_{v,0}, \boldsymbol{x}_{v,1}, \boldsymbol{x}_{v,2}, \dots, \boldsymbol{x}_{v,K_v})\}$ be a set of sequences (with different lengths $K_v \leq K$) of vectors in a (possibly uncountable) set of features $\mathcal{X} \subset \mathbb{R}^d$. Consider a diagonal linear complex-valued RNN with $d_s$-dimensional hidden state, parameters $\boldsymbol{\Lambda} \in diag(\mathbb{C}^{d_s})$, $\boldsymbol{W}_{in} \in \mathbb{C}^{d \times d_s}$ and recurrence rule $\boldsymbol{s}_{v,k} = \boldsymbol{\Lambda} \boldsymbol{s}_{v,k-1} + \boldsymbol{W}_{in} \boldsymbol{x}_{v,K_v-k}$, initialized at $\boldsymbol{s}_{v,-1} = \boldsymbol{0} \in \mathbb{R}^{d_s}$ for all $v \in V$. If $d_s \geq Kd$, then there exist $\boldsymbol{\Lambda}, \boldsymbol{W}_{in}$ such that the map $R : (\boldsymbol{x}_{v,0}, \boldsymbol{x}_{v,1}, \boldsymbol{x}_{v,2}, \dots, \boldsymbol{x}_{v,K}) \mapsto \boldsymbol{s}_{v,K}$ (with zero right-padding if $K_v < K$) is bijective. Moreover, if the set of RNN inputs has countable cardinality $|\mathcal{X}| = N \leq \infty$, then selecting $d_s \geq d$ is sufficient for the existence of an injective linear RNN mapping $R$.*

*Proof.* For now, let us assume for ease of exposition that all sequences are of length $K$, coinciding with the graph diameter. The extension to the general setting is trivial and presented at the end of the proof as a separate paragraph. Also, let us, for simplicity, drop the dependency on $v \in V$ and talk about generic sequences.

The proof simply relies on the idea of writing the linear recurrence in matrix form (Orvieto et al., 2023a; Gu et al., 2022). Note that for a generic input $\boldsymbol{x} = (\boldsymbol{x}_0, \boldsymbol{x}_1, \boldsymbol{x}_2, \dots, \boldsymbol{x}_K) \in \mathbb{R}^{d \times (K+1)}$, the recurrence output can be rewritten in terms of powers of $\boldsymbol{\Lambda} = \text{diag}(\lambda_1, \lambda_2, \dots, \lambda_{d_s})$ as follows (see Sec. 4):

$$\boldsymbol{s}_K = \sum_{k=0}^{K} \boldsymbol{\Lambda}^k \boldsymbol{W}_{in} \boldsymbol{x}_k. \tag{8}$$

We now present sufficient conditions for the map $R : (\boldsymbol{x}_0 \boldsymbol{x}_1, \boldsymbol{x}_2, \dots, \boldsymbol{x}_K) \mapsto \boldsymbol{s}_K$ to be injective or bijective. The proof for bijectivity does not require the set of node features to be in a countable set, and it is simpler.

**Bijective mapping.**    First, let us design a proper matrix $\boldsymbol{W}_{in} \in \mathbb{R}^{d_s \times d}$. We choose $d_s = (K+1)d$ and set $\boldsymbol{W}_{in} = \boldsymbol{I}_{d \times d} \otimes \boldsymbol{1}_{(K+1) \times 1}$. As a result, the RNN will independently process each sequence dimension with a sub-RNN of size $K$. The resulting $\boldsymbol{s}_K \in \mathbb{R}^{(K+1)d}$ will gather each sub-RNN output by concatenation. We can then restrict our attention to the first components of the input sequence.

$$(\boldsymbol{s}_K)_{1:(K+1)} = \sum_{k=0}^{K} \text{diag}(\lambda_1, \lambda_2, \dots, \lambda_{d_s})^k \boldsymbol{1}_{(K+1) \times 1} x_{k,1}. \tag{9}$$

This sum can be written conveniently with multiplications using a Vandermonde matrix:

$$(\boldsymbol{s}_L)_{1:(K+1)} = \begin{pmatrix} \lambda_1^K & \lambda_1^{K-1} & \cdots & \lambda_1 & 1 \\ \lambda_2^K & \lambda_2^{K-1} & \cdots & \lambda_2 & 1 \\ \vdots & \vdots & \ddots & \vdots & \vdots \\ \lambda_{K+1}^K & \lambda_{K+1}^{K-1} & \cdots & \lambda_K & 1 \end{pmatrix} \boldsymbol{x}_{0:K,1}^{\leftarrow}. \tag{10}$$

where $\boldsymbol{x}_{0:K,1}^{\leftarrow}$ is the input with reversed arrow of time. The proof is concluded by noting that Vandermonde matrices of size $(K+1) \times (K+1)$ are full-rank since they have non-zero determinant $\prod_{1 \leq i < j \leq d_s} (\lambda_i - \lambda_j) \neq 0$, under the assumption that all $\lambda_i$ are distinct. Note that one does not need complex eigenvalues to achieve this, both $\boldsymbol{\Lambda}$ and $\boldsymbol{W}_{in}$ are real. However, as discussed in (Orvieto et al., 2023a), complex eigenvalues improve conditioning of the Vandermonde matrix.

**Injective mapping.**    The condition for injectivity is that if $\boldsymbol{x} \neq \hat{\boldsymbol{x}}$, then $R(\boldsymbol{x}) \neq R(\hat{\boldsymbol{x}})$. In formulas,

$$\boldsymbol{s}_K - \hat{\boldsymbol{s}}_K = \sum_{k=0}^{K} \boldsymbol{\Lambda}^k \boldsymbol{W}_{in} (\boldsymbol{x}_k - \hat{\boldsymbol{x}}_k) \neq 0 \tag{11}$$

Let us assume the state dimension coincides with the input dimension, and let us set $\boldsymbol{W}_{in} = \boldsymbol{I}_{d \times d}$. Then, we have the condition

$$\boldsymbol{s}_K - \hat{\boldsymbol{s}}_K = \sum_{k=0}^{K} \boldsymbol{\Lambda}^k (\boldsymbol{x}_k - \hat{\boldsymbol{x}}_k) \neq 0. \tag{12}$$

Since $\mathbf{\Lambda} = \mathrm{diag}(\lambda_1, \lambda_2, \ldots, \lambda_{d_s})$ is diagonal, we can study each component of $\boldsymbol{s}_K - \hat{\boldsymbol{s}}_K$ separately. We therefore require

$$s_{K,i} - \hat{s}_{K,i} = \sum_{k=0}^{K} \lambda_i^k (x_{k,i} - \hat{x}_{k,i}) \neq 0 \qquad \forall i \in \{1, 2, \ldots, d\}. \tag{13}$$

We can then restrict our attention to linear one-dimensional RNNs (i.e. filters) with one-dimensional input $\boldsymbol{x} \in \mathbb{R}^{1 \times (K+1)}$. We would like to choose $\lambda \in \mathbb{C}$ such that

$$\sum_{k=0}^{K} \lambda^k (x_k - \hat{x}_k) \neq 0 \tag{14}$$

Under the assumption $|\mathcal{X}| = N \leq \infty$, $\boldsymbol{x} - \bar{\boldsymbol{x}}$ is a generic signal in a countable set ($N(N-1)/2 = \Omega(N^2)$ possible choices). Let us rename $\boldsymbol{z} := \boldsymbol{x} - \bar{\boldsymbol{x}} \in \mathcal{Z} \subset \mathbb{R}^{1 \times (K+1)}, |\mathcal{Z}| = \Omega(N^2)$. We need

$$\langle \bar{\boldsymbol{\lambda}}, \boldsymbol{z} \rangle \neq 0, \qquad \forall \boldsymbol{z} \in \mathcal{Z}, \qquad \text{where} \quad \bar{\boldsymbol{\lambda}} = (1, \lambda, \lambda^2, \cdots, \lambda^K) \tag{15}$$

Such $\lambda$ can always be found *in the real numbers*, and the reason is purely geometric. We need

$$\bar{\boldsymbol{\lambda}} \notin \mathcal{Z}_\perp := \bigcup_{\boldsymbol{z} \in \mathcal{Z}} \boldsymbol{z}_\perp.$$

Note that $\dim(\boldsymbol{z}_\perp) = K$, so $\dim(\mathcal{Z}_\perp) = K$ due to the countability assumption — in other words the Lebesgue measure vanishes: $\mu(\mathcal{Z}_\perp; \mathbb{R}^{K+1}) = 0$. If $\bar{\boldsymbol{\lambda}}$ were an arbitrary vector, we would be done since we can pick it at random and with probability one $\bar{\boldsymbol{\lambda}} \notin \mathcal{Z}_\perp$. But $\bar{\boldsymbol{\lambda}}$ is structured (lives on a 1-dimensional manifold), so we need one additional step.

Note that $\bar{\boldsymbol{\lambda}}$ is parametrized by $\lambda$, and in particular $\mathbb{R} \ni \lambda \mapsto \bar{\boldsymbol{\lambda}} \in \mathbb{R}^{K+1}$ is a curve in $\mathbb{R}^{K+1}$, we denote this as $\gamma_\lambda$. Now, crucially, note that the support of $\gamma_\lambda$ is a smooth curved manifold for $K > 1$. In addition, crucially, $0 \notin \gamma_\lambda$. We are done: it is impossible for the $\gamma_\lambda$ curve to live in a $K$ dimensional space composed of a union of hyperplanes: it indeed has to span to whole $\mathbb{R}^{K+1}$, without touching the zero vector (see Figure 4). The reason why it spans the whole $\mathbb{R}^{K+1}$ comes from the Vandermonde determinant! Let $\{\lambda_1, \lambda_2, \cdots, \lambda_K\}$ be a set of $K$ distinct $\lambda$ values. The Vandermonde matrix

$$\begin{pmatrix} \lambda_1^K & \lambda_1^{K-1} & \cdots & \lambda_1 & 1 \\ \lambda_2^K & \lambda_2^{K-2} & \cdots & \lambda_2 & 1 \\ \vdots & \vdots & \ddots & \vdots & \vdots \\ \lambda_K^{K+1} & \lambda_K^{K-2} & \cdots & \lambda_K & 1 \end{pmatrix}$$

has determinant $\prod_{1 \leq i < j \leq d_s}(\lambda_i - \lambda_j) \neq 0$ — its full rank, meaning that the vectors $\bar{\boldsymbol{\lambda}}_1, \bar{\boldsymbol{\lambda}}_2, \ldots, \bar{\boldsymbol{\lambda}}_{K+1}$ span the whole $\mathbb{R}^{K+1}$. note that $\lambda \mapsto \bar{\boldsymbol{\lambda}}$ is a continuous function, so even though the single $\bar{\boldsymbol{\lambda}}_i$ might live on $\mathcal{Z}_\perp$ there exist a value in between them which is not contained in $\mathcal{Z}_\perp$.

$\square$

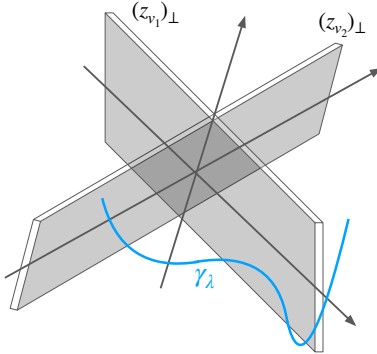

Figure 4: Proof illustration for Proposition 4.1 (injectivity proof). The set $\mathcal{Z}_\perp$ is depicted as union of hyperplanes, living in $\mathbb{R}^{K+1}$ and here sketched in three dimensions. The curve $\gamma_\gamma : \gamma \mapsto (1, \gamma, \gamma^2, \cdots, \gamma^K)$ is shown as a blue line. The proof shows that, for $\gamma \in \mathbb{R}$, the support of $\gamma_\lambda$ is not entirely contained in $\mathcal{Z}_\perp$.

# B  SIMPLIFIED IMPLEMENTATION OF ONE GRED LAYER

```python
import jax.numpy as jnp
import flax.linen as nn
from typing import Callable

class MLP(nn.Module):
    dim_h: int
    drop_rate: float = 0.

    @nn.compact
    def __call__(self, inputs, training: bool = False):
        x = nn.LayerNorm()(inputs)
        x = nn.Dense(self.dim_h)(x)
        x = nn.gelu(x)
        x = nn.Dropout(self.drop_rate, deterministic=not training)(x)
        x = nn.Dense(self.dim_h)(x)
        x = nn.Dropout(self.drop_rate, deterministic=not training)(x)
        return x + inputs

def binary_operator_diag(element_i, element_j):
    # Binary operator for parallel scan of linear recurrence.
    a_i, bu_i = element_i
    a_j, bu_j = element_j
    return a_j * a_i, bu_i * a_j + bu_j

class LRU(nn.Module):
    dim_v: int # State dimension
    dim_h: int
    drop_rate: float
    init_eigenvalue_magnitude: Callable
    init_eigenvalue_phase: Callable
    init_kernel: Callable

    @nn.compact
    def __call__(self, inputs, training: bool = False):
        # Shape of inputs: (K+1, batch_size, num_nodes, dim_h)
        xs = nn.LayerNorm()(inputs)

        # Construct Lambda:
        nu_log = self.param("nu_log", self.init_eigenvalue_magnitude, (self.dim_v,))
        theta_log = self.param("theta_log", self.init_eigenvalue_phase, (self.dim_v,))
        diag_lambda = jnp.exp(-jnp.exp(nu_log) + 1j * jnp.exp(theta_log))

        # Construct W_in:
        W_in_re = self.param("W_in_re", self.init_kernel, (self.dim_h, self.dim_v))
        W_in_im = self.param("W_in_im", self.init_kernel, (self.dim_h, self.dim_v))
        W_in = W_in_re + 1j * W_in_im

        # Parallel scan over the sequence of length K+1:
        xs = xs @ W_in
        lambdas = jnp.repeat(diag_lambda[None, ...], inputs.shape[0], axis=0)
        lambdas = jnp.expand_dims(lambdas, axis=(1, 2))
```

```
52          _, xs = jax.lax.associative_scan(binary_operator_diag, (lambdas, xs), reverse=True)
53          x = xs[0]
54
55          # Project the complex-valued hidden state to real:
56          W_out_re = self.param("W_out_re", self.init_kernel, (self.dim_v, self.dim_h))
57          W_out_im = self.param("W_out_im", self.init_kernel, (self.dim_v, self.dim_h))
58          W_out = W_out_re + 1j * W_out_im
59          x = nn.gelu((x @ W_out).real)
60
61          # Apply GLU:
62          x = nn.Dropout(self.drop_rate, deterministic=not training)(x)
63          x = nn.Dense(self.dim_h)(x) * jax.nn.sigmoid(nn.Dense(self.dim_h)(x))
64          x = nn.Dropout(self.drop_rate, deterministic=not training)(x)
65          return x + inputs[0]
66
67 class GRED(nn.Module):
68     dim_v: int # State dimension
69     dim_h: int
70     drop_rate: float
71     init_eigenvalue_magnitude: Callable
72     init_eigenvalue_phase: Callable
73     init_kernel: Callable
74
75     @nn.compact
76     def __call__(self, inputs, dist_masks, training: bool = False):
77          # Shape of inputs: (batch_size, num_nodes, dim_h)
78          # Shape of dist_masks: (batch_size, K+1, num_nodes, num_nodes)
79          xs = jnp.swapaxes(dist_masks, 0, 1) @ inputs
80          xs = MLP(self.dim_h, self.drop_rate)(xs)
81          x = LRU(
82              self.dim_v,
83              self.dim_h,
84              self.drop_rate,
85              self.init_eigenvalue_magnitude,
86              self.init_eigenvalue_phase,
87              self.init_kernel
88          )(xs, training=training)
89          return x
```

## C  HYPERPARAMETERS

Table 5: Hyperparameters for GRED. "-" indicates that $K$ is the diameter of the graph.

| Hyperparameter | ZINC 12K | MNIST | CIFAR10 | PATTERN | CLUSTER | Peptides |
|:---:|:---:|:---:|:---:|:---:|:---:|:---:|
| # Layers | 11 | 4 | 8 | 10 | 16 | 8 |
| $K$ | 4 | 2 | 4 | - | - | 40 |
| Dropout | 0.2 | 0.15 | 0.15 | 0.2 | 0.2 | 0.2 |
| $d$ | 72 | 128 | 96 | 64 | 64 | 88 |
| $d_s$ | 72 | 128 | 64 | 64 | 64 | 88 |
| Learning rate | 0.001 | 0.001 | 0.001 | 0.001 | 0.001 | 0.001 |
| Weight decay | 0.1 | 0.1 | 0.1 | 0.1 | 0.1 | 0.2 |
| # Epochs | 2000 | 200 | 200 | 100 | 100 | 200 |
| Batch size | 32 | 16 | 16 | 32 | 32 | 32 |

# D ADDITIONAL EXPERIMENTS

**Effect of $K$ on performance**  We show how different $K$ values affect the performance of GRED on CIFAR10 (Table 6), ZINC (Table 7) and Peptides-func (Table 8). We use $K_{\max}$ to denote the maximum diameter of all graphs in the dataset. For Peptides-func, the maximum $K$ we tried was smaller than $K_{\max}$ in order to fit the model into a single RTX A5000 GPU with 24GB memory. From the three tables, we can observe that larger $K$ values generally yield better performance. On CIFAR10 and ZINC, while directly using $K_{\max}$ already outperforms MPNNs, the optimal value of $K$ yielding best performance lies strictly between 1 and $K_{\max}$. This may be because information that is too far away is less important for these two tasks (interestingly, the best $K$ value for CIFAR10 is similar to the width of a convolutional kernel on a normal image). On Peptides-func, the change of performance is more monotonic in $K$. When $K = 40$, GRED outperforms the best graph transformer GRIT. This result is impressive considering that GRED doesn't use any positional encoding, and further validates that the architecture of GRED alone can encode the graph structure. We observe no further performance gain when we increase $K$ to 60.

Table 6: Effect of $K$ on the performance on CIFAR10.

| $K$ | 1 | 4 | 7 | $K_{\max}$=10 |
|---|---|---|---|---|
| Test Acc (%) | 72.540$\pm$0.336 | **75.370$\pm$0.621** | 74.490$\pm$0.335 | 74.210$\pm$0.274 |

Table 7: Effect of $K$ on the performance on ZINC.

| $K$ | 1 | 2 | 4 | 8 | $K_{\max}$=22 |
|---|---|---|---|---|---|
| Test MAE $\downarrow$ | 0.231$\pm$0.002 | 0.161$\pm$0.003 | **0.089$\pm$0.004** | 0.108$\pm$0.004 | 0.131$\pm$0.008 |

Table 8: Effect of $K$ on the performance on Peptides-func.

| $K$ | 5 | 10 | 20 | 40 | 60 |
|---|---|---|---|---|---|
| Test AP $\uparrow$ | 0.6657$\pm$0.0069 | 0.6883$\pm$0.0076 | 0.6960$\pm$0.0060 | **0.7041$\pm$0.0049** | 0.7031$\pm$0.0017 |

**Vanilla RNN vs LRU**  We replace the LRU component (5) of GRED with a vanilla RNN:

$$\boldsymbol{s}_{v,k}^{(\ell)} = \tanh\left(\boldsymbol{W}_{\mathrm{rec}}\boldsymbol{s}_{v,k-1}^{(\ell)} + \boldsymbol{W}_{\mathrm{in}}\boldsymbol{x}_{v,K-k}^{(\ell)}\right), \tag{16}$$

where $\boldsymbol{W}_{\mathrm{rec}} \in \mathbb{R}^{d_s \times d_s}$ and $\boldsymbol{W}_{\mathrm{in}} \in \mathbb{R}^{d_s \times d}$ are two trainable real weight matrices, and show the difference in performance in Table 9. We use the same number of layers and the same $K$ for both models. We can observe that the performance of GRED with a vanilla RNN drops significantly. On CIFAR10 and ZINC where $K$ is small, GRED$_{\mathrm{RNN}}$ still outperforms the best MPNN. However, on Peptides-func where we use 8 layers and $K = 40$ per layer, the vanilla RNN becomes difficult to train and the performance of GRED$_{\mathrm{RNN}}$ is even worse than the best MPNN.

Table 9: Performance of GRED using vanilla RNN or LRU.

| | CIFAR10 $\uparrow$ | ZINC $\downarrow$ | Peptides-func $\uparrow$ |
|---|---|---|---|
| Best MPNN | 67.312$\pm$0.311 | 0.188$\pm$0.004 | 0.6069$\pm$0.0035 |
| GRED$_{\mathrm{RNN}}$ | 69.215$\pm$0.080 | 0.160$\pm$0.005 | 0.4945$\pm$0.0024 |
| GRED$_{\mathrm{LRU}}$ | **75.370$\pm$0.621** | **0.089$\pm$0.004** | **0.7041$\pm$0.0049** |

**Performance on TUDataset**  We further evaluate GRED on NCI1 and PROTEINS from TU-Dataset. We follow the experimental setup of Abboud et al. (2022), and report the average accuracy and standard deviation of 10 splits, as shown in Table 10. Our model generalizes well to TUDataset and shows good performance. Furthermore, GRED outperforms SPN (Abboud et al., 2022) with the same number of hops, which validates that GRED is a more effective architecture for utilizing information from a large neighborhood.

Table 10: Performance (accuracy) of GRED on TUDataset.

| Model | NCI1 | PROTEINS |
|---|---|---|
| DGCNN | 76.4±1.7 | 72.9±3.5 |
| DiffPool | 76.9±1.9 | 73.7±3.5 |
| ECC | 76.2±1.4 | 72.3±3.4 |
| GIN | 80.0±1.4 | 73.3±4.0 |
| GraphSAGE | 76.0±1.8 | 73.0±4.5 |
| SPN (Abboud et al., 2022) ($K = 10$) | 78.2±1.2 | 74.5±3.2 |
| GRED ($K = 10$) | **82.6±1.4** | **75.0±2.9** |

## E    COMPUTATIONAL GRAPHS OF MPNN AND GRED

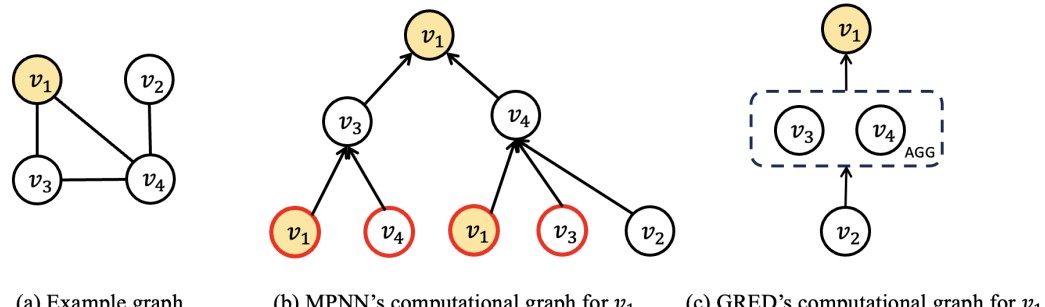

(a) Example graph        (b) MPNN's computational graph for $v_1$        (c) GRED's computational graph for $v_1$

Figure 5: We compare the computational graphs of MPNN and GRED that generate the representation for the same target node $v_1$ in the graph (a). For MPNN to access all nodes in the graph, 2 rounds of message passing are needed. As a result, the computational graph of MPNN (b) includes duplicate nodes (red circles). On the contrary, the computational graph of GRED (c) encodes each node exactly once, and GRED relies on the inductive bias of recurrence to encode the distance information.

