# OpenReview forum: "Recurrent Distance-Encoding Neural Networks for Graph Representation Learning"
_ICLR.cc/2024/Conference — Submitted to ICLR 2024_

### Official Review · Reviewer_hzhG · 2023-10-24

**Soundness:** 3 good
**Presentation:** 3 good
**Contribution:** 2 fair
**Rating:** 6
**Confidence:** 4

**Summary:**

The paper proposed a new architecture for the graph learning, aggregating nodes with different distance and using a parallelizable linear recurrent network to encode the information flow while keeping the weights from vanishing.

**Strengths:**

1. paper is well written and easy to follow
2. experiment is good

**Weaknesses:**

1. RNN here is strange, the assumption here is that the information flow from K-hop to K-1 hop. but information can also go "outward", from k-2 hop to k-1 hop. RNN is not good to model the graph information
2. I assume all the GNN drawback will also be here in this Model (over smoothing, etc). The model can be regarded as another interpretation of massage passing(even though the weight is different), for the target node, the model will aggregate all the K-hop node information

**Questions:**

1. how the performance like if we change LRU to the vanilla RNN. seems LRU for the long range information pass is the key factor to win against MPNN.

---

> ### Author Response · Authors · 2023-11-21
> **Response to Reviewer hzhG**
>
> We thank reviewer hzhG for taking the time to review our work, and are pleased she/he found our work to be well-written and our experiments to be good. We address the raised concerns as follows:
>
> **Q1. RNN here is strange, the assumption here is that the information flow from K-hop to K-1 hop. but information can also go "outward", from k-2 hop to k-1 hop. RNN is not good to model the graph information**.
>
> **Ans**: It’s true that each edge in the graph allows “bidirectional” information flow. However, if we focus on a particular node for which we want to generate a representation, it’s natural to categorize the other nodes into “shells” (Figure 1(b)). Nodes that belong to the same shell have the same shortest distance from the target node (considering running a breadth-first search from the target node). Then a recurrent neural network, which runs from the outermost shell to the innermost one, would be a reasonable choice to encode the neighborhood structure.
>
> Besides, our model indeed allows “bidirectional” information flow. For two different target nodes (Node 1 and Node 2), their shells are different, the K-1 hop and K hop neighbors of Node 1 can become the K hop and K-1 hop respectively for Node 2, and the information propagates reversely when we generate the representation for Node 2. Therefore, considering the set of node representations as a whole, we won’t lose any structural information. This argument is supported by our rigorous theoretical analysis of expressiveness
>
> **Q2. I assume all the GNN drawback will also be here in this Model (over smoothing,  etc). The model can be regarded as another interpretation of massage passing(even though the weight is different), for the target node, the model will aggregate all the K-hop node information**.
>
> **Ans**: GRED doesn’t have the drawbacks of MPNN because the computational graphs of GRED and MPNN are different. We have added a straightforward example to Appendix E of the updated PDF to illustrate their difference. To aggregate all nodes in the K-hop neighborhood, MPNN needs to perform message passing for K rounds. As a result,  MPNN’s computational graph for the target node grows exponentially with K because the same neighbor will appear many times in the computational graph. The exponentially growing number of nodes are mixed together by MPNN to generate the representation for the target node, resulting in the known drawbacks (e.g., over-smoothing, over-squashing).  On the contrary, the computational graph of GRED includes each node in the K-hop neighborhood exactly once, and GRED further utilizes LRU to control how signals of distant nodes are propagated toward the target node. Therefore, GRED can more effectively leverage information beyond the local neighborhood than MPNN.
>
> **Q3. how the performance like if we change LRU to the vanilla RNN. seems LRU for the long range information pass is the key factor to win against MPNN**.
>
> **Ans**: We have conducted experiments to compare LRU with vanilla RNN:
>
> | ~   |  CIFAR10 $\uparrow$ | ZINC $\downarrow$ | Peptides-func $\uparrow$ |
> | --- | --- | --- | --- |
> |    Best MPNN      | 67.312$\pm$0.311 | 0.188$\pm$0.004 | 0.6069$\pm$0.0035   |
> | GRED$_{\text{RNN}}$   | 69.215$\pm$0.080 | 0.160$\pm$0.005 | 0.4945$\pm$0.0024 |
> | GRED$_{\text{LRU}}$   | **75.370$\pm$0.621** | **0.089$\pm$0.004** | **0.7041$\pm$0.0049** |
>
> We replace the LRU component of GRED with a vanilla RNN:
>
> \begin{equation}
>     \mathbf{s}_{v, k}^{(l)} = \text{tanh} ( \mathbf{W}\_{\text{rec}} \mathbf{s}\_{v, k - 1}^{(l)} + \mathbf{W}\_{\text{in}}\mathbf{x}\_{v, K - k}^{(l)})
> \end{equation}
>
> where $\mathbf{W}\_{\text{rec}} \in \mathbb{R}^{d_s \times d_s}$ and $\mathbf{W}\_{\text{in}} \in \mathbb{R}^{d_s \times d}$ are two trainable real weight matrices. We use the same number of layers and the same $K$ for both models. We can observe that the performance of GRED with a vanilla RNN drops significantly. On CIFAR10 and ZINC where $K$ is not large, GRED$\_{\text{RNN}}$ still outperforms the best MPNN, but on Peptides-func where long-range information is needed (8 layers with $K=40$ per layer), the performance of GRED$\_{\text{RNN}}$ is worse than the best MPNN because vanilla RNNs are known to be difficult to train on long sequences, unlike the LRU.
>
> We kindly ask the reviewer to consider raising her/his score if she/he thinks our response can address the concerns. We would be happy to answer further questions!

---

> > ### Comment · Reviewer_hzhG · 2023-11-22
> > **thanks for the clarification**
> >
> > thank you for the answers to my questions. My questions have been clarified. I'll raise my rating to weak accept.

---

### Official Review · Reviewer_8Mos · 2023-10-30

**Soundness:** 3 good
**Presentation:** 3 good
**Contribution:** 2 fair
**Rating:** 3
**Confidence:** 4

**Summary:**

The paper presents a new graph learning architecture called Graph Recurrent Encoding by Distance (GRED). GRED aims to address the challenges faced by existing approaches, such as message-passing neural networks (MPNNs) and graph transformers, by efficiently encoding information from distant nodes while avoiding the need for ad-hoc positional encodings. The paper provides a detailed explanation of the GRED architecture and supports its claims with theoretical analysis and empirical results.

**Strengths:**

1. Theoretical Analysis: The paper provides a theoretical analysis of the expressiveness of GRED, demonstrating its superiority over one-hop MPNNs. This analysis adds depth to the understanding of the method's capabilities. Furthermore, the authors provide an interesting analysis regarding the RNN filtering of the k-hop neighbor features.

2. The presentation and clarity of the paper are very good. The authors provide comprehensive explanations of the key components of GRED, which is crucial for readers trying to understand the architecture at a deeper level. The theoretical analysis is rigorous, and the empirical results are presented in a well-organized manner, contributing to a comprehensive understanding of GRED's performance.

**Weaknesses:**

1. Firstly, the novelty of the proposed architecture is somewhat limited, as there are already several similar approaches in the field that operate on K-hop neighborhoods in a similar manner. While the combination of permutation-invariant neural networks and linear recurrent networks is a sensible choice, it may not present a significant departure from existing methods. Specifically, the proposed method is very similar to [1], which proposed the following update rule: $h_u^{(t+1)} =COM(h_u^t, AGG_{u,1},..., AGG_{u,k} )$. The main difference is that the proposed approach uses an RNN for the $COM$ function. Moreover, there is no proper discussion of other k-hop approaches such as [2].

2. Secondly, the experiments in the paper are conducted on a relatively limited set of datasets, focusing on long-rage benchmarks, and the absence of experiments on benchmark datasets like TUDatasets raises concerns about the generalizability of the proposed model. Expanding the experimental evaluation to a wider range of datasets would strengthen the paper's claims.

3. A notable weak point of the paper is the unavailability of the source code for the proposed model. This omission hinders the reproducibility and transparency of the research. Releasing the source code is not only a common practice in the research community but is also crucial for enabling other researchers to validate and build upon the work presented in the paper.

References:

[1] Abboud, Ralph, Radoslav Dimitrov, and Ismail Ilkan Ceylan. "Shortest path networks for graph property prediction." Learning on Graphs Conference. PMLR, 2022.

[2] Nikolentzos, Giannis, George Dasoulas, and Michalis Vazirgiannis. "k-hop graph neural networks." Neural Networks 130 (2020): 195-205.

**Questions:**

1. Could the authors provide a more in-depth comparative analysis of their approach against existing methods that operate on k-hop neighborhoods? Highlighting specific strengths or weaknesses in comparison to these related approaches would help better position the novelty of this work.

2. Considering that experiments are conducted on a limited set of datasets, could the authors discuss the generalizability of their proposed architecture to a wider range of datasets, including TUDatasets?

---

> ### Author Response · Authors · 2023-11-21
> **Response to Reviewer 8Mos**
>
> We thank reviewer 8Mos for taking the time to review our work and are pleased she/he found: 1) our presentation to be very good | clear; 2) our theory to be interesting | rigorous; and 3) our empirical results to be well-organized and contributing to a comprehensive understanding of GRED's performance.
>
> We admit that we missed several references on K-hop GNNs, and we agree that a more detailed discussion of these methods would help improve the understanding of our contributions. We address these concerns as follows:
>
> **Q1. The novelty of the proposed architecture is somewhat limited | Lack of a comparative analysis of existing K-hop GNNs**.
>
> **Ans**: Compared with existing K-hop GNNs, the novelty of our proposed architecture is a new way to encode the K-hop neighborhood, i.e., with LRU. **The use of LRU ensures our model theoretically sound expressiveness as well as ability to address the over-squashing problem in long-range reasoning tasks**. To see why the novelty of our model is significant, we provide a comparative analysis of existing K-hop GNNs here:
>
> Among K-hop GNNs, MixHop [3] uses powers of the adjacency matrix to access k-hop nodes. k-hop GNN [2] iteratively applies MLPs to combine two consecutive hops and propagates information towards the target node. **While they have shown that higher hop information can improve the expressiveness of MPNNs, they suffer from over-squashing**, i.e., they cannot effectively utilize information from distant nodes because the signals of distant nodes would decay rapidly due to iterative neighborhood mixing. As a result, the largest neighborhood these approaches operate on is 3-hop. The drawback of these approaches is also pointed out by the SPN paper [1]. The SPN paper [1] proposes a conceptual update formula which is similar to ours:
>
> $$
> h\_u^{(t+1)} = \text{COM} ( h\_u^{(t)}, \text{AGG}\_{u, 1}, …, \text{AGG}\_{u, K})
> $$
>
> **However, SPN simply uses weighted summation as COM to aggregate K hops. The weighted summation cannot guarantee the expressiveness of the model since it cannot uniquely represent sequences. On the contrary, the LRU component of GRED can achieve expressiveness and long-range modeling capacity simultaneously**: LRU can express an injective mapping of sequences, and at the same time prevent distant information from decaying by learning proper eigenvalues of the transition matrix.
> We have updated the related work to include a comparison of K-hop GNNs (marked in red in the updated PDF)
>
> To validate that GRED can encode the K-hop neighborhood more effectively than SPN,
> We evaluate GRED on two additional datasets from TUDataset and compare against SPN.
> As shown in Part III of “Additional Experimental Results”, GRED can outperform SPN with the same number of hops.
>
> [1] Shortest Path Networks for Graph Property Prediction. Learning on Graphs Conference. 2022
>
> [2] k-hop graph neural networks. Neural Networks. 2020
>
> [3] MixHop: Higher-Order Graph Convolutional Architectures via Sparsified Neighborhood Mixing. ICML. 2019

---

> > ### Author Response · Authors · 2023-11-21
> > **Response to Reviewer 8Mos (Continued)**
> >
> > **Q2. Considering that experiments are conducted on a limited set of datasets, could the authors discuss the generalizability of their proposed architecture to a wider range of datasets, including TUDatasets?**
> >
> > **Ans**: We evaluated our model on 7 datasets in total. They come from different domains (images, social networks, biology), cover both long-range and non-long-range tasks, and are also widely used by graph transformer baselines we compare against. **Both Reviewer e7jX and Reviewer bs5m agree that our experiments are extensive**. To quote Reviewer e7jX: ‘the evaluation is very thorough’ and ‘the proposed method is benchmarked against a wide range of methods and on multiple benchmarks’, and to quote Reviewer bs5m: “The paper compares the proposed method on multiple benchmarks, and demonstrates its effectiveness against traditional GNN models”.
> >
> > We further evaluate GRED on two additional datasets from TUDataset and compare against SPN. We follow the experimental setup of SPN, and report the average accuracy and standard deviation of $10$ splits:
> >
> > |Model |  NCI1          | PROTEINS |
> > | --- | --- | --- |
> > |DGCNN |  76.4$\pm$1.7  | 72.9$\pm$3.5 |
> > |DiffPool | 76.9$\pm$1.9 | 73.7$\pm$3.5 |
> > |ECC      | 76.2$\pm$1.4 | 72.3$\pm$3.4 |
> > |GIN      | 80.0$\pm$1.4 | 73.3$\pm$4.0 |
> > |GraphSAGE | 76.0$\pm$1.8 | 73.0$\pm$4.5 |
> > |SPN [1] ($K=10$) | 78.2$\pm$1.2 | 74.5$\pm$3.2 |
> > |GRED ($K=10$) | **82.6$\pm$1.4** | **75.0$\pm$2.9** |
> >
> > The baseline performance is directly copied from the SPN paper. GRED generalizes well to TUDataset and outperforms SPN with the same number of hops.
> >
> > **Q3. A notable weak point of the paper is the unavailability of the source code for the proposed model**.
> >
> > **Ans**: We have added an implementation of GRED to Appendix B of the updated PDF. We will release the full training scripts after the paper is accepted.
> >
> > We kindly ask the reviewer to consider raising her/his score if she/he thinks our response can address the concerns. We would be happy to answer further questions!

---

> > > ### Author Response · Authors · 2023-11-23
> > > **Kind Reminder**
> > >
> > > Dear Reviewer 8Mos,
> > >
> > > Thank you once again for your time spent reviewing our paper! In our response, we have managed to address all the raised concerns. The discussion period is ending soon, and we would really appreciate it if you could provide feedback. We are happy to answer any further questions!
> > >
> > > Best regards,
> > > The authors

---

### Official Review · Reviewer_bs5m · 2023-11-01

**Soundness:** 3 good
**Presentation:** 3 good
**Contribution:** 3 good
**Rating:** 6
**Confidence:** 3

**Summary:**

The paper proposes a model architecture, GRED, for graph tasks by adopting a recurrent neural network (RNN) to propagate hidden states through multiple layers. It claims that the new architecture is more effective, utilizing the information of large neighborhoods with high efficiency, and theoretically proving the expressiveness. It also empirically shows that the performance of GRED is better than state-of-the-art graph transformers.

**Strengths:**

1.  The paper proposes a way to utilize RNN to adopt hidden states through multiple layers. It claims that the new method helps improve long-range information processing.
2. The paper compares the proposed method on multiple benchmarks, and demonstrates its effectiveness against traditional GNN models.
3. The description of the proposed method is well-written and easy to understand, though figure 1(a) is a bit unclear about what the "skip" stands for.

**Weaknesses:**

1. For the training efficiency evaluation, it would be nice to include the memory consumption for different methods. Also, the performance gap on the task between GRIT and GRED is still significant, which makes the comparison a bit unfair. It would be better to compare with the architecture that has similar task performance.
2. The method introduces a new hyperparameter K to be tuned. It would be nice to show how K is selected, and how the different selections of K can affect the task performance.
3. Although the technique helps increase the range of nodes the model can process, unlike the graph transformer-based method, the distance is still limited by the selection of K and the number of layers of the model. It would be good to show some insights into how that super long-distance information affects the model performance.

**Questions:**

The following papers seem related as well:
1. Graph Transformers: Representing Long-Range Context for Graph Neural Networks with Global Attention, NeurIPS 2021
2. Issues with attention for long-range reasoning: Lite Transformer with Long-Short Range Attention, ICLR 2020

---

> ### Author Response · Authors · 2023-11-21
> **Response to Reviewer bs5m**
>
> We thank reviewer bs5m for taking the time to review our work and are pleased she/he found our work to be well-written and easy to understand.  First, we would like to ask the reviewer to take a look at our post “Additional Experimental Results” for updated results and the sensitivity analysis with respect to the value of $K$. We address the raised concerns as follows:
>
> **Q1. What is the "skip" in Figure 1(a)?**
>
> **Ans**: The “skip” in Figure 1(a) denotes an identity branch. We have added a code snippet of GRED to Appendix B of the updated PDF, to show how the computation is done exactly.
>
> **Q2. For the training efficiency evaluation, it would be nice to include the memory consumption for different methods. Also, the performance gap on the task between GRIT and GRED is still significant, which makes the comparison a bit unfair. It would be better to compare with the architecture that has similar task performance.**
>
> **Ans**: We have added the GPU memory consumption to Table 4 of the updated PDF. We also post it here:
>
> | Model | ZINC 12K     | CIFAR10 | Peptides-func |
> | --- | --- | --- | --- |
> | GRIT  | 25.6s / 1.9GB   | 244.4s / 4.6GB  | 225.6s / 22.5GB |
> | GRED (Ours)  | 4.1s / 1.4GB  | 27.8s / 1.4GB  | 158.9s / 18.5GB  |
> | Speedup | **6.2$\times$**| **8.8$\times$** |  **1.4$\times$** |
>
> The gap between the performance of GRIT and the updated performance of GRED is much less significant now. Especially on Peptides-func with $K=40$ GRED can outperform GRIT. GRED still maintains higher training efficiency than GRIT.
>
>
> **Q3. The method introduces a new hyperparameter K to be tuned. It would be nice to show how K is selected, and how the different selections of K can affect the task performance.**
>
> **Ans**: We have conducted experiments on how different $K$ values affect the performance of GRED. Please refer to Part I of our “Additional Experimental Results” thread in our general response for a thorough analysis.
>
> **Q4. Although the technique helps increase the range of nodes the model can process, unlike the graph transformer-based method, the distance is still limited by the selection of K and the number of layers of the model. It would be good to show some insights into how that super long-distance information affects the model performance.**
>
> **Ans**: It’s true that the range of nodes each layer can access is bounded by the value of $K$. However, for real-world datasets, the graph diameter won’t increase linearly with the number of nodes. Therefore, even if the value of $K$ doesn’t seem “super long”, each layer of our model can access all nodes in the graph.
>
> For all datasets we used in experiments, except for Peptides-func and Peptides-struct, our model fits into a single 24GB GPU even when $K$ is set to $K_{\text{max}}$ (i.e., the maximum diameter of all graphs in the dataset). For Peptides-func and Peptides-struct, we use a $K$ value which is smaller than $K_{\text{max}}$ in order to increase the depth of the model. In this sense, the choice of $K$ allows the model some flexibility when the hardware resource is constrained. In all cases, we ensure that model depth * $K$ is strictly larger than $K_{\text{max}}$, so the receptive field of the entire GRED model is the same as graph transformers and GRED is able to extract global information.
>
> **Q5. Two related papers**
>
> **Ans**: We have added these two references to the related work (marked in red) of the updated PDF.
>
> We kindly ask the reviewer to consider raising her/his score if she/he thinks our response can address the concerns. We would be happy to answer further questions!

---

> > ### Author Response · Authors · 2023-11-23
> > **Kind Reminder**
> >
> > Dear Reviewer bs5m,
> >
> > Thank you once again for your time spent reviewing our paper! In our response, we have managed to address all the raised concerns. The discussion period is ending soon, and we would really appreciate it if you could provide feedback. We are happy to answer any further questions!
> >
> > Best regards,
> > The authors

---

> > ### Comment · Reviewer_bs5m · 2023-11-23
> > **RE: Response to Reviewer bs5m**
> >
> > Thanks for the reply! The response from the authors solved most of my concerns. I increased the score to 6.

---

### Official Review · Reviewer_e7jX · 2023-11-01

**Soundness:** 4 excellent
**Presentation:** 3 good
**Contribution:** 4 excellent
**Rating:** 6
**Confidence:** 3

**Summary:**

This work introduces a novel graph learning method that pools long-range information by aggregating nodes at different distances and using a linear recurrent network, leading to a computational efficient method that is competitive with state-of-the-art approaches for graph learning.

**Strengths:**

- The evaluation is very thorough and the results validate and support the effectiveness of the proposed method. The computational effectiveness of the proposed model compared to equivalent transformer models (Table 4) is very good.
- The proposed method is benchmarked against a wide range of methods and on multiple benchmarks.
- Transformer models are good at modeling long-range interactions in graphs, and have been outperforming MPNN models, they however struggle with scaling and require custom encoding of the node positional embedding. The proposed approach is simple and intuitive and provides an alternative path towards solving long-range graph problems.

**Weaknesses:**

1. A sensitivity analysis of GRED with respect to the choice of K is missing.
2. The training efficiency analysis is conducted against a transformer model only, but it would be useful to understand how this method compares to MPNNs. In particular, each node will have its own sequence of sets of nodes at distance k, so there might not be any shared computation that can be leveraged like in iterative 1-hop message passing methods.

**Questions:**

- How would this method perform on non long-range benchmarks? Would it underperform compared to MPNNs who might only need a local receptive field to solve a task?

---

> ### Author Response · Authors · 2023-11-21
> **Response to Reviewer e7jX**
>
> We thank reviewer e7jX for taking the time to review our work, and we are pleased she/he found our evaluation to be very thorough and our results to validate the effectiveness of our model. We address the raised concerns as follows:
>
> **Q1. A sensitivity analysis of GRED with respect to the choice of K is missing**.
>
> **Ans**: This is now in Appendix D of the updated PDF. Please refer to Part I of our “Additional Experimental Results” thread in our general response for a thorough analysis.
>
> **Q2. The training efficiency analysis is conducted against a transformer model only, …, each node will have its own sequence of sets of nodes at distance k, so there might not be any shared computation that can be leveraged…**
>
> **Ans**: It’s true that each node has its own sequence of set representations, but the computations for all nodes in the graph can be done in batches. In the implementation, we use a matrix mask to indicate each node’s $k$-hop neighbors (just like the adjacency matrix indicates each node’s 1-hop neighbors). These masks are computed only once in preprocessing using the shortest path algorithm, and they are used to do aggregation at each hop. Then the input to LRU would be a batch of sequences of length K+1 (including the target node for each sequence). **We have added a code snippet of GRED in Appendix B of the updated PDF, to show how the computation is done**.
>
> To compare the training efficiency of GRED with MPNN, we use the same number of layers but the MPNN counterpart only aggregates 1-hop neighbors at each layer. We measure the average training time per epoch:
>
> | Model | ZINC | CIFAR10 | Peptides-func |
> | --- | --- | --- | --- |
> | GRED | 4.1s | 27.8s | 158.9s |
> | MPNN | 2.8s | 21.7s | 91.6s |
>
> We can observe that on ZINC and CIFAR10, the additional time required to use a larger neighborhood is marginal. On Peptides-func, the difference is greater since GRED uses $K=40$. However, this extra time, which is still less than the time required by GRIT, is acceptable considering the significant performance improvement
>
> **Q3. How would this method perform on non long-range benchmarks? Would it underperform compared to MPNNs who might only need a local receptive field to solve a task?**
>
> **Ans**: The datasets in Table 1 and Table 2 (ZINC) are from [4], which is a widely used benchmark. These datasets are not particularly long-range. PATTERN and CLUSTER model communities in social networks and all nodes can be reached within 3 hops. MNIST and CIFAR10 are nearest-neighbor graphs of pixels, and just like their normal versions local information is important. Our model performs well on these datasets. We further evaluate our model on two additional datasets from TUDataset (please refer to Part III of our “Additional Experimental Results” thread) and our model also achieves good performance.
>
> Our model can adapt to both long-range and non-long-range tasks by learning proper eigenvalues of the LRU transition matrix that control how fast the filters decay as the distance increases. For a long-range task, the learned eigenvalues are close to 1 which better keeps distant information, while for a non-long-range task, the magnitudes of the learned eigenvalues are smaller (please refer to Figure 3 for an illustration of the eigenvalues after training). Additionally, the choice of $K$ gives our model some flexibility to adjust the receptive field, and we found the optimal $K$ for a non-long-range task lies between 1 and the diameter (Part I of “Additional Experimental Results”)
>
> [4] Benchmarking Graph Neural Networks. JMLR. 2022
>
> We kindly ask the reviewer to consider raising her/his score if she/he thinks our response can address the concerns. We would be happy to answer further questions!

---

> > ### Author Response · Authors · 2023-11-23
> > **Kind Reminder**
> >
> > Dear Reviewer e7jX,
> >
> > Thank you once again for your time spent reviewing our paper! In our response, we have managed to address all the raised concerns. The discussion period is ending soon, and we would really appreciate it if you could provide feedback. We are happy to answer any further questions!
> >
> > Best regards,
> > The authors

---

### Author Response · Authors · 2023-11-21
**General Response from Authors**

We thank the reviewers for their interesting questions and constructive feedback. **We incorporated in the paper revision all of the additional experiments and ablations requested by the reviewers**. We summarize our additional experimental results in the thread below. We are very happy with these new results that further strengthen our paper. If the reviewers also believe we addressed all of their concerns and curiosities, we kindly ask them to revise their scores.

---

> ### Author Response · Authors · 2023-11-21
> **Additional Experimental Results**
>
> In this thread, we present additional experimental results requested by the reviewers. In part I, we show how different $K$ values (number of hops in the recurrence) affect the performance of our model. In part II, we replace the LRU component with a vanilla RNN and show the difference in performance (confirming the effectiveness of our LRU choice). In part III, we evaluate our model on two additional datasets from TUDataset. Our model generalizes well to TUDataset and outperforms another related approach SPN [1]. All the above results have been added to Appendix D of the updated PDF.
>
> Moreover, with further hyper-parameter tuning, we have improved the performance of our model on CIFAR10 and Peptides-func: for CIFAR10 we increase the model depth with a smaller hidden dimension, and for Peptides-func we increase $K$ to 40. The hyper-parameters for our model are summarized in Appendix C. **Especially, when $K=40$ the performance of GRED on Peptides-func (0.7041±0.0049) is better than the best graph transformer GRIT (0.6988±0.0082), and GRED still maintains higher efficiency than GRIT (158.9s/epoch vs 225.6s/epoch)**. This new result is impressive given that GRED doesn’t use any positional encoding and further validates that the architecture of GRED alone can encode the structural information.
>
> **Part I: Effect of $K$ on performance**
>
> Effect of $K$ on the performance on CIFAR10:
> |       $K$           |        1                       |         4             |       7        |      $K_{\text{max}}$=10   |
> | --- | --- | --- | --- | --- |
> | Test Acc (\%)  |  72.540$\pm$0.336 | **75.370$\pm$0.621** |  74.490$\pm$0.335 | 74.210$\pm$0.274 |
>
> Effect of $K$ on the performance on ZINC:
> |     $K$          |  1 | 2 | 4 | 8 | $K_{\text{max}}$=22 |
> | --- | --- | --- | --- | --- | --- |
> |    Test MAE $\downarrow$   |  0.231$\pm$0.002 | 0.161$\pm$0.003 |  **0.089$\pm$0.004** | 0.108$\pm$0.004 | 0.131$\pm$0.008 |
>
> Effect of $K$ on the performance on Peptides-func:
> | $K$               | 5 | 10 | 20 | 40 | 60 |
> | --- | --- | --- | --- | --- | --- |
> | Test AP $\uparrow$   | 0.6657$\pm$0.0069 | 0.6883$\pm$0.0076 | 0.6960$\pm$0.0060 | **0.7041$\pm$0.0049** | 0.7031$\pm$0.0017 |
>
> We use $K_{\text{max}}$ to denote the maximum diameter of all graphs in the dataset. For Peptides-func, the maximum $K$ we tried was slightly smaller than $K_{\text{max}}$ in order to fit the model into a single RTX A5000 GPU with 24GB memory. From the three tables, we can observe that larger $K$ values generally yield better performance. On CIFAR10 and ZINC, while directly using $K_{\text{max}}$ already outperforms MPNNs, the optimal value of $K$ yielding best performance lies strictly between $1$ and $K_{\text{max}}$. We think this is because information that is too far away is less important for these two tasks: indeed, interestingly, the best $K$ value for CIFAR10 is similar to the width of a convolutional kernel on a normal image. On Peptides-func, the change of performance is more monotonic in $K$. When $K=40$, GRED outperforms the best graph transformer GRIT. We observe no further performance gain when we increase $K$ to 60.

---

> > ### Author Response · Authors · 2023-11-21
> > **Additional Experimental Results (Continued)**
> >
> > **Part II: Vanilla RNN vs LRU**
> >
> > Performance of GRED using vanilla RNN or LRU:
> > | ~   |  CIFAR10 $\uparrow$ | ZINC $\downarrow$ | Peptides-func $\uparrow$ |
> > | --- | --- | --- | --- |
> > |    Best MPNN      | 67.312$\pm$0.311 | 0.188$\pm$0.004 | 0.6069$\pm$0.0035   |
> > | GRED$_{\text{RNN}}$   | 69.215$\pm$0.080 | 0.160$\pm$0.005 | 0.4945$\pm$0.0024 |
> > | GRED$_{\text{LRU}}$   | **75.370$\pm$0.621** | **0.089$\pm$0.004** | **0.7041$\pm$0.0049** |
> >
> > We replace the LRU component of GRED with a vanilla RNN:
> >
> > \begin{equation}
> >     \mathbf{s}_{v, k}^{(l)} = \text{tanh} ( \mathbf{W}\_{\text{rec}} \mathbf{s}\_{v, k - 1}^{(l)} + \mathbf{W}\_{\text{in}}\mathbf{x}\_{v, K - k}^{(l)})
> > \end{equation}
> >
> > where $\mathbf{W}\_{\text{rec}} \in \mathbb{R}^{d_s \times d_s}$ and $\mathbf{W}\_{\text{in}} \in \mathbb{R}^{d_s \times d}$ are two trainable real weight matrices. We use the same number of layers and the same $K$ for both models. We can observe that the performance of GRED with a vanilla RNN drops significantly. On CIFAR10 and ZINC where $K$ is not large, GRED$\_{\text{RNN}}$ still outperforms the best MPNN. However, on the long-range dataset Peptides-func where we use $8$ layers and $K=40$ per layer, the vanilla RNN becomes difficult to train, and the performance of GRED$_{\text{RNN}}$ is even worse than the best MPNN.
> >
> > **Part III: Performance on TUDataset**
> >
> > We further evaluate GRED on NCI1 and PROTEINS from TUDataset.
> > We follow the experimental setup of [1], and report the average accuracy and standard deviation of $10$ splits:
> >
> > |Model |  NCI1          | PROTEINS |
> > | --- | --- | --- |
> > |DGCNN |  76.4$\pm$1.7  | 72.9$\pm$3.5 |
> > |DiffPool | 76.9$\pm$1.9 | 73.7$\pm$3.5 |
> > |ECC      | 76.2$\pm$1.4 | 72.3$\pm$3.4 |
> > |GIN      | 80.0$\pm$1.4 | 73.3$\pm$4.0 |
> > |GraphSAGE | 76.0$\pm$1.8 | 73.0$\pm$4.5 |
> > |SPN [1] ($K=10$) | 78.2$\pm$1.2 | 74.5$\pm$3.2 |
> > |GRED ($K=10$) | **82.6$\pm$1.4** | **75.0$\pm$2.9** |
> >
> > The baseline performance is directly copied from [1]. Our model (GRED) shows very good performance. In particular, GRED outperforms SPN [1] with the same number of hops, which validates that GRED is a more effective architecture than SPN for utilizing information beyond the local neighborhood.
> >
> > [1] Shortest Path Networks for Graph Property Prediction. Learning on Graphs Conference. 2022

---

### Meta-Review · Area_Chair_3SvH · 2023-12-10

**Metareview:**

This work introduces a graph learning method, GRED, which aggregates nodes at different shortest path distances and uses a linear recurrent network to combine them. The reviewers have reached a weak consensus for acceptance. However, a rejection is still recommended due to its limited novelty. The authors have addressed its connection with some previous work in their response to Q1 of Reviewer 8Mos. However, one important work, kP-GNN [Feng et al., 2022], has been overlooked by the authors in both the response and the related work section in the paper. As stated in Section 4.2 of the paper, GRED in this work is a specific implementation of K-hop message passing in [Feng et al., 2022]. The only novelty of GRED is the usage of a linear RNN, which is equivalent to using a linear layer with shared parameters as the UPD function in Definition 1. Compared with the original implementation (MLP as the UPD), GRED does not bring extra expressivity or theoretical time/space complexity reduction. Furthermore, the motivation for using a linear RNN is scalability. However, a scalability comparison between [Feng et al., 2022] and GRED is not available at this time. The theoretical analysis also seems to be largel built upon [Feng et al., 2022].

**Justification For Why Not Higher Score:**

One important previous work [Feng et al., 2022] (cited in the paper) seems to be systematically downplayed. Despite much of the paper's methodology and theory are built upon [Feng et al., 2022], the paper unfortunately does not even discuss it in the related work.

**Justification For Why Not Lower Score:**

N/A

---

### Decision · Program_Chairs · 2024-01-16

Reject